# Investigation of Perceived Stress During COVID-19 Pandemic Self-Isolation Periods

**DOI:** 10.3390/medicina61020175

**Published:** 2025-01-21

**Authors:** Paulius Ūselis, Živilė Jacikė, Audronė Šeibokaitė, Aušra Griciūtė

**Affiliations:** 1Department of Health Psychology, Faculty of Public Health, Academy of Medicine, Lithuanian University of Health Sciences, LT-44307 Kaunas, Lithuania; ausra.griciute@lsmuni.lt; 2Public Institution Kaunas City Outpatients Clinic, Pramones pr. 31, LT-51270 Kaunas, Lithuania; zivile.jacike@kaunopoliklinika.lt (Ž.J.); audrone.seibokaite@kaunopoliklinika.lt (A.Š.)

**Keywords:** self-isolation, psychological examination, laboratory stress indicators

## Abstract

*Background and Objectives*: The study purpose was to analyze possible health consequences of self-isolation during the COVID-19 pandemic, aiming to evaluate diagnostics methods. Specifically, we analyzed perceived stress of self-isolation with the aim of evaluating the suitability of psychological and laboratory diagnostics methods for routine clinical practice. In order to achieve the aim of the study, the following objectives were formulated: to compare the results of psychological and laboratory diagnostic methods between case and control groups; and to evaluate associations between psychological and laboratory stress indicators separately in case and control groups. *Materials and Methods*: The research study consisted of control and case groups of 28 volunteers each. The main selection criterion for the case group was self-isolation due to COVID-19 and a maximum period of 3 months after post-isolation, while the control group had to be of a similar age but did not have to be isolated or self-isolated. Both groups consisted of young (18–24 years) individuals. All participants had to fill out a Perceived Stress Scale (PSS) questionnaire and were subjected to a laboratory test for stress indicators (alpha-amylase, secretory cortisol, and immunoglobulin A) from a saliva sample. *Results*: A comparison of the laboratory stress indicator scores for both study groups revealed statistically significant differences between the clinical subgroups, i.e., the distributions of the control and case groups were significantly different within the affected case group and control. The values obtained for study groups and PSS scores showed no discrepancies between the two investigation methods, i.e., PSS assessment and laboratory stress indicators results. The PSS values between the clinical groups were significantly different from each other, suggesting that the laboratory stress indicator scores differed but were consistent or complementary to the PSS results. A separate comparison of age and stress indicator levels in the control group revealed a correlation between age and PSS scores, indicating that younger individuals were more prone to subjective perception of moderate stress. *Conclusions*: The results showed that COVID-19 self-isolation during quarantine affected people’s psychological health. Using psychological examination and laboratory stress indicators, the results of the case group reliably differed from the results of the control group, allowing us to conclude that self-isolation more often caused moderate chronic stress, with or without decompensation. Besides the main study objective, we observed that laboratory stress biomarkers may be acceptable for broader clinical application during routine psychological treatment. The clinical application of laboratory stress biomarkers had been validated previously by another method, i.e., psychological investigation using PSS.

## 1. Introduction

The COVID-19 pandemic, which began in late autumn 2019, marked a threat to humanity and will go down in history as one of the worst pandemics, claiming nearly seven million lives and sickening and infecting hundreds of millions with the SARS-CoV-2 virus, according to John Hopkins University data from an interactive web-based dashboard to track COVID-19 in real time [1]. As the COVID-19 pandemic continued, various restrictions and bans continued globally, despite the push for vaccination, leaving governments to enact more or less restrictive measures, with subsequent disturbances of daily routines and stability. The resulting stressful conditions are having long-lasting psychological consequences, which will be the focus of future investigations [2].

Modern humans have little knowledge of the effects of isolation or self-isolation or its necessity in an age of rapid travel, constant movement, and intercommunication. The terms “isolation” and “self-isolation” have quite similar meanings in epidemiology guidelines, i.e., for people who have or are suspected of having COVID-19 to avoid passing the infection to other people [3]. The term “social isolation” covers all precautions related to the pandemic and its consequences on mental health [2]. Although the extents of previous outbreaks of Severe Acute Respiratory Syndrome (SARS), Middle East Respiratory Syndrome (MERS), Spanish flu, Ebola, and plague shocked the world, the dominance of technology was not as great as it is today [4]. The sudden catastrophic turn of events forced humanity to confront a painful realization—how to live with itself. Paradoxically, the “virtual connection” provided by social media has made us forget how close relationships should feel, how we feel when we cannot touch our relatives and friends, when we cannot say hello, shake hands, hug, or cuddle. Fear, submissiveness, uncertainty, denial, and aggression—a myriad of emotions that have come at once—have created a plethora of psychological challenges that need to be tackled [4].

According to a previous meta-analysis, a lack of social connections increases health risks as much as smoking 15 cigarettes a day or having an alcohol drinking problem. Scientists also found that loneliness and social isolation are twice as harmful to physical and mental health as obesity [5].

The duration of quarantine, fear of infection, frustration, boredom, multiple interpretations of social phenomena, a lack of information, financial loss, and stigma [6]—these concepts were daily companions for people living in quarantine or in self-isolation. Negative psychological effects during quarantine are not surprising, but there is growing evidence that these effects can be detected months or years later as a reactive state of the body, although there is not enough research in this direction to date [6].

Data published by researchers on the psychological distress that occurs during quarantine, and the lack of data on the changes in personal and social life and their impact on the health of the body in the face of the COVID-19 pandemic, have led to the need for a comprehensive analysis of the psychological state of people in self-isolation. Analyzing the mental health consequences, the most popular and well know method is the “Perceived Stress Scale” (PSS), first used and described in 1983 by Cohen and colleagues [7]. The PSS is widely considered as the gold standard for measuring stress perception. An individual’s response to a stressor is sometimes multifaceted, especially regarding the impact of the stressor on physical health, e.g., when caring for a family member who is elderly, illness is often considered a chronic stressor because it evokes constant physical and emotional strain [8]. 

Classical biomarkers of stress include endocrine changes, in particular, the metabolic cycling of hormones such as cortisol and epinephrine [9,10]. It is the hypothalamic-pituitary-adrenal system, along with the autonomic nervous and immune systems, that become sensitive and immediately responsive to changes in peripheral stress by releasing widely known stress biomarkers such as cortisol, alpha amylase, and anti-inflammatory cytokines [9,10]. Stress markers can be classified into diagnostic, prognostic, and therapeutic types, according to their application. Diagnostic ones are those that aid in the diagnosis or detection of a disease. Prognostic markers help to predict the course of a disease. Therapeutic markers are used to monitor treatment and prevent disease progression [11]. In a general sense, secretory cortisol, alpha amylase, and immunoglobulin tests are standards in clinical practice to assess whether the body is showing a nonspecific response to a stressor. The obtained results are always interpreted by a clinician, with conclusions or indications being specific to obtained data [12]. The laboratory of the Kaunas City Polyclinic was the first in Lithuania to introduce tests for secretory stress indicators (cortisol, alpha-amylase, and secretory immunoglobulin) from saliva samples of patients in 2018. The the test is gradually finding its way to practical application, helping physicians to diagnose a wide range of pathologic conditions and to objectively monitor courses of treatment [13].

Testing all three stress indicators, i.e., secretory Cortisol (sC), secretory α-Amylase (sAA), and secretory immunoglobulin A (sIgA), simultaneously and assessing their changes can answer questions related to the nature of stress (i.e., is it acute or chronic), to adaptation to stress (is it adaptive or exhaustive), and to its severity or the response to it, as well as in the detection of endocrine (adrenal hypo- or hyper-function) or neurological (autonomic dysfunction) pathology, etc. [13]. Psychological scales and laboratory stress indicator methodologies expand the scope of comprehensive analyses of the effects of several groups of stressors on the body, such as a person’s fear of contracting an as-yet-unknown or unstudied, dangerous infection and the consequences of a period of psychological discomfort in the context of self-isolation.

## 2. Materials and Methods

The present case-control study was conducted in the context of the COVID-19 pandemic, and therefore, personal protective equipment (medical masks, disposable laboratory gloves when collecting saliva samples from respondents) and a distance of 1.5 m during direct contact were observed. The study was conducted in accordance with the declaration Helsinki and its protocol was approved by the Lithuanian Bioethics Committee (No BEC-SP(B)-69). On 31 January 2022, an approval letter was obtained for this study. Informed consent was obtained from all subjects involved in the study in written form. The study procedures were approved by the Chief Executive Officer of the Public Institution Kaunas City Outpatients Clinic.

The selection criteria for the case group were as follows: residence in of Kaunas city, aged 18–24 years, who needed to be in self-isolation due to COVID-19 not longer than 3 months before our testing was undertaken. The control group was composed of individuals of the same age but who had not undergone any type of isolation. The case and control groups consisted of 56 people, with 28 in each group. The structure of the study is presented in Figure 1. All participant data were de-personalized in the follow-up study, in accordance with ethical requirements and confidentiality.

The collection of the test samples was carried out over a period of 6 months, from December 2021 to May 2022, with visits to the homes of the respondents and saliva sampling from those who came to the laboratory reception at the Kaunas city Outpatient Clinic, Dainava branch. Respondents signed an informed consent agreement to participate in the study, providing their age, sex, and duration and end date of self-isolation, and completed a test based on the Perceived Stress Scale (PSS) method, known as a traditional test to assess how different stressful situations affect feelings [14]. The number of recruited subjects for the case group was 30, but two of them were not assessed with the stress indicator test due to pre-analysis errors (insufficient saliva in the sample or food residues in the saliva), resulting in a total sample size of 28 subjects, an equal number to that in the control group.

Using the results of the PSS survey, the participants were grouped into three categories according to their scores: scores between 0 and 13, considered to reveal a low level of stress; scores between 14 and 26, indicating moderate stress; and scores between 27 and 40, with high stress stratification [7].

Saliva samples for laboratory testing for stress indicators were collected upon completion of a questionnaire, using the laboratory’s approved saliva collection manual, in the presence of a qualified biomedical technologist. The samples were then tested in the Kaunas City clinic laboratory according to the approved ELISA methods using a Euroimmun analyzer from EUROIMMUN Medical Diagnostics (Beijing, China) Ltd., supplied by local distributor “Dimuna”, Kaunas. The obtained results were evaluated by a panel of three laboratory staff: a qualified biomedical technologist (BMT), responsible for the proper collection and preparation of the sample; a qualified BMT, who performed the immunoenzymatic analyses of the collected samples; and a laboratory doctor, who evaluated the quality of the performed tests and provided authorization and conclusions for each test according to validated and approved procedures. After a final assessment, the stress indicator results were grouped into standardized categories, resulting in three main groups, in line with the Clinical Group System (CGS), based on key information from the stress indicators reagent inserts (Instructions for use, accessed on 12 January 2019) [15,16,17].

CSI I. Healthy and experiencing moderate daily stress.CSI II.   Chronic stress with or without decompensation.CSI III.  Predominant somatic symptoms, possibility of compensated stress.

The results of the quantitative study (Psychological Assessment (PA)) and the laboratory results were subjected to statistical analysis using SPSS Statistics V.23 software (IBM, New York, NY, USA). The Kolmogorov-Smirnov test was used to test the distribution of the data. Continuous variables were expressed as mean and standard deviations when they did not deviate from normal distribution, or as median and minimum and maximum values when they deviated reliably from normal distribution. Qualitative variables were expressed as absolute numbers and percentages.

Differences in the means of the quantitative variables between the two study groups were compared using the Student *t*-test. ANOVA test with LSD post hoc criterion was used to compare the three CGS subgroups. When the distribution differed from normal, the Mann Whitney test was used was used for group comparisons. Spearman’s correlation (Spearman rho) was used to determine correlations among data. Chi-square (*χ*^2^) was used to test for associations between qualitative variables.

The association between variables was considered statistically significant at *p* < 0.05.

## 3. Results

### 3.1. Study Demographics

Two groups of respondents were recruited: case, *N* = 28, and control, *N* = 28. They were grouped by age and gender and all of them underwent PSS and laboratory stress indicators tests. After assessing the subjects according to gender and age, no statistically significant differences were found between the groups for either criterion (Table 1).

### 3.2. Assessment of Correlations Between PSS Scores in Case and Control Groups

A comparison of the subjective perceived stress (PSS) scores between the case and control groups showed no statistically significant differences, with t = 1.017 *p* = 0.314.

Comparing the frequencies of laboratory stress indicator scores in the clinical groups (CGS), significant differences were found between the groups (Figure 2), i.e., the distributions of the control and case groups were significantly different within the groups. There was a pronounced inverse distribution of results in both groups. In the control group, 20 subjects were classified as CGS I, i.e., healthy people or those with symptoms of mild daily stress. In contrast, the case group was dominated by 20 cases of CGS II, i.e., characterized by symptoms of chronic long-term stress with or without decompensation. An equal number of people in both the case and control groups were categorized as CGS III, i.e., predominantly somatic illnesses or symptoms of compensated stress. Therefore, 70% of the control group was healthy or under mild stress, while only 10% of the case group was healthy. Meanwhile, 70% of the case group showed chronic stress with or without decompensation; this figure was only 10% in the control group. The same numbers of people with somatic disorders or under the influence of medication were found in both groups. Statistical differences between the three CGS groups were obtained by assessing the results, without distinguishing between the case and the control group separately and comparing their CGS results with each other. Figure 2 shows a graphical representation of the data.

### 3.3. Assessment of Laboratory Stress Indicators and Clinical Groups in Both Study Groups 

A comparison of the stress marker median results between the case and the control groups showed statistical differences between the groups (*p* ≤ 0.05) regarding sC and sIgA, while the results of sAA did not differ between the groups (*p* = 0.210) (Table 2).

A comparison of the PSS scores (Figure 3) according to the interval-specific values, i.e., PSS scores (scores between 0 and 13 are considered low stress, scores between 14 and 26 are considered moderate stress; scores between 27 and 40 belong to high stress sequence) and a comparison of the scores obtained with the equivalent groups of CGS showed statistically significant differences between CGS I and CGS II and between CGS II and CGS III, with *p* ≤ 0.05. This assessment contributes to our observation that the two applied methodologies are complementary, i.e.,: in CGS group I, healthy or mild-moderate stress (PSS mean value 24.43); in CGS group II, moderate (PSS mean value 20.48) long-term stress with or without decompensation; and in CGS group III, predominantly somatic pathology, with the possible influence of medication, causes mild stress (PSS mean value 13.20). Graphical representation of the mean scores with data intervals shows a clear distribution of data (Figure 3).

### 3.4. Analysis of Laboratory Stress Indicators, PSS Sores, and Age in the Case and Control Groups

Our evaluation of mean age and stress indicator concentrations in the control group showed a negative correlation between age and PSS scores, with younger age groups having higher PSS scores, sAA negatively correlated with sC concentrations, and sIgA negatively correlated with sC concentrations. Meanwhile, no difference was observed in the case group. Only sIgA and sAA were positively correlated in the subject group (Table 3 and Table 4).

## 4. Discussion

The results showed that COVID-19 self-isolation during quarantine affects people’s psychological health. In a recent Lancet article, Brooks and colleagues [18] suggested that even short-term isolation of up to 10 days can lead to long-term psychiatric symptomatology, with residual post-stress effects possible after even three years or more [19]. Thus, our choice of a maximum period of three months after self-isolation did not contradict the researchers’ opinion regarding the validity of this criterion for detecting possible stress-related sequelae. When assessing self-isolation during the period under study, stressors (forced dissociation, fear of infection, lack of information or persistent negative information, uncertainty, etc.) acting on the body were found to lead to a continuous cascade of consequences [20,21]. Therefore, by using not only psychological scales, but also specific stress indicators—laboratory markers—in our study, a more comprehensive clinical picture could be produced to help structure psychological support and to unveil new comprehensive testing methodologies [12,22].

First of all, the two groups were considered similar, with equal numbers of men and women in the case and control groups (39%). The distribution by sex and age (18–24 years) did not show any statistical demographic differences between groups. The relatively small sample size (28 subjects and the same number in the control group) may suggest an under-representation of respondents, but similar samples sizes (27 respondents each) were recently collected by Croatian researchers [23] in a study conducted and published on healthy students in different faculties (Kinesiology and mixed faculty group), assessing the body’s reactions to daily stress, comparing the results of analogous stress indicators (sC, sAA) as in our work, and providing a perceived stress scale (PSS) assessment. The published work showed a reliable statistical correlation between age and secretory alpha-amylase levels in saliva, depending on gender in both groups, and a significant difference in the male students at the Faculty of Kinesiology. The direct correlation between secretory cortisol and secretory alpha amylase showed a positive statistical correlation, i.e., as one indicator decreased, so did the other.

Similar work by Polish researchers also used the PSS methodology and the same stress indicators as in our work, i.e., sC and sAA [22]. A large cohort study was conducted over a period of three years on young Polish school-aged children, divided into two groups (6–7 and 8–11 years), with a total of 260 boys and 243 girls. More than half of the children had been exposed to stressful situations in the past two weeks. The results showed that salivary alpha-amylase was negatively correlated with the occurrence of psychological stress in the PSS scores, i.e., lower levels of the marker were found in the children who were stressed than in those who were not. No differences were found for secretory cortisol and secretory immunoglobulin. Thus, when assessing respondents of different age groups, the influence of hormones on the results could certainly be the main reason [22,24]. The effect and the body’s response are different in children, different in young people, and even different in older and elderly age groups, as stress hormones are also influenced by physical activity during the day [23]. Therefore, our choice of the age range and the relative equality of the two groups may have eliminated the influence of age as a potential confounder of the results.

Our study involved young people, 40% of whom were students in the Lithuanian population. All subjects were exposed to the same COVID-19 self-isolation situation: more than 10 days of COVID-19 quarantine-related isolation in a home environment. However, when considering the COVID-19 situation itself, both in Lithuania and worldwide, and considering it as a long-term stressor for the population as a whole, the control group was equally exposed to this stressor, with the only difference being that in this case, they did not experience self-isolation, which is to be considered as an exceptional criterion for the evaluation of our results. This may have been a factor in the significance of the PSS results, as no statistical differences were obtained when comparing the groups of respondents on the basis of the psychological scales alone. However, after statistical analysis and a comparison of the laboratory stress indicator scores according to the CGS, i.e., grouping them together, calling them clinical groups for the sake of clarity, and standardizing them from I to III, where, respectively, those who were in group I were healthy, did not experience any stress, or experienced only short-term daily stress, and the other groups were differentiated accordingly (see Methods of data collection, evaluation and analysis), a correlation in the PSS scores was observed by comparing the laboratory stress indicators among the CGS groups. The most frequently observed chronic stress (20 subjects, 3 controls) was in the moderately perceived stress formulation. Statistical differences were found when comparing the means of the PSS according to the CGS. The first group of CGS (healthy or daily stress) had a mean PSS of 24.43, which is typical of moderate stress (PSS value range 14–26), as well as chronic stress CGS II with a value of 20.48, which is characteristic of subjects in the range of moderate stress values. Therefore, it can be assumed that self-isolation acted as a stressor and that the juxtaposition of the two different methodologies added to the overall results.

Interesting results were obtained in the presence of somatic pathology, a possible pain syndrome, compensated chronic stress, or medication to compensate for the pathological effects on the body. We can only make assumptions regarding compounding effects as possible but not disclosed influences among our study subjects. Five subjects each from the case group and the control group fell into the same third clinical group, while the PSS scores for the same subjects showed a mean score of 13.2, characteristic of a low level of stress. Thus, five people in the control group may have been experiencing low perceived stress due to somatic, internal organ pathology, whereas, equally, five people in the control group, even in the presence of possible internal organ pathology, were experiencing lower perceived stress in a different, self-isolating context. One could consider which characteristic predominated, i.e., psychological discomfort or the stress of somatic pathology, as the main determinant of differences within such a broad clinical group determination [13]. Here, a larger sample would be needed to obtain deeper insights.

Meanwhile, statistically significant differences were found regarding secretory immunoglobulin A and secretory Cortisol when comparing the mean results for the two study groups. Secretory alpha-amylase concentrations showed no statistical difference between both groups. Hence, when exposed to prolonged stress, amplified by the self-isolation, the body responds and activates the sympathetic-adrenal-medullary (SAM) system and the hypothalamic-pituitary-adrenal (HPA) axis, which produces and secretes specific hormones [12,24]. Therefore, a pronounced psychological response to emotional stress also modulates immune functions, the autonomic nervous system (ANS), hypothalamic-pituitary hormones, neuropeptides, cytokines, and other factors associated with this network. Thus, short- and long-term exposure to stress is associated with changes in the functioning of the pituitary–pituitary axis [25]. Factors that alter glucocorticoid levels begin to predominate and may affect different outcomes. The stress response involves functions between the central nervous system (CNS) and the immune system, with bidirectional connections. Furthermore, adaptation to stress is a crucial mechanism of the body’s response to stress, where the effectiveness of the stress response depends on the type, intensity, and duration of the stress, as well as the characteristics of the individual [8]. Therefore, it is very likely that there was no statistical significance in the change in alpha amylase levels when comparing our subjects and the control group, i.e., when the predominant effect of prolonged stress was enhanced by additional self-isolation stimuli, stimulated adrenal activity (increase in secretory cortisol levels in the subject group), and a feedback mechanism via the hypothalamus [26], causing changes in the immune system such as an increase in secretory immunoglobulin in the subject group. A physiological response to SARS-CoV-2 virus cannot be ruled out, as immunoglobulin A is known to be a secretory marker of a non-inflammatory process in the mucosa in the presence of pathogens [12]. As the subjects were not asked about other possible infections in the present, this possibility cannot be excluded, as self-isolation was already applied in the event of a positive COVID-19 test or illness in a member of the close environment. This assumption was reinforced by the significant statistical difference in secretory immunoglobulin A levels in the control group.

Some secretory biomarkers (cortisol, alpha-amylase, pro-inflammatory cytokines) serve as stress biomarkers, reflecting both ANS and HPA activity. Among the various factors belonging to the neuroendocrine axis, cortisol plays a crucial role in the stress response. Many works in the literature justify the role of secretory cortisol in the diagnosis of diseases [26], i.e., in assessing somatic conditions or distinguishing purely psychological causes. Thus, in our work, the concentration of secretory alpha-amylase did not differ in the study and control groups, and the statistical differences found in secretory cortisol and secretory immunoglobulin A concentrations may be explained by pathophysiological mechanisms. We assume that chronic stress can also weaken the immune response; as evidenced by the study of antibody responses to vaccines, it can cause or contribute to various diseases such as cardiovascular, endocrine, gastrointestinal diseases, and related conditions. The scientific literature indicates that severe stress can also alter the levels of different immune factors by increasing them [20]. Increased cortisol levels lead to HPA hyperactivity, thereby increasing the risk of various diseases [26]. Meanwhile, alpha-amylase, which reflects the state of the autonomic nervous system and, depending on the context of the study, i.e., whether there was only acute stress due to somatic illness or whether pharmacological agents were used, can be applied, in principle, to study ANS activity [27,28]. Alpha-amylase may therefore be a suitable biomarker for the assessment of the ANS in the context of behavioral medicine. The results of a Croatian student study, where alpha-amylase activity was found to be higher in students at the Faculty of Kinesiology, who were probably engaged in physical activities that triggered a cascade of neurohumoral mechanisms related to acute physiological stress, supports this hypothesis.

Advantages of the study: a complex examination was applied, combining the methods of psychological examination (PSS) and laboratory diagnostics, i.e., laboratory stress markers were tested by a clinical method. This is the first work in Lithuania to investigate the psychological consequences of self-isolation during COVID-19 quarantine for people aged 18–24 using PSS and laboratory stress indicators, and therefore, the first study to establish the clinical correspondence of PSS and laboratory stress indicator values, confirming the possibility of applying two different methods that complement each other in practice.

Limitations: This study used a relatively small sample size, and additional clinical insights could be expected by increasing sample sizes. It is possible that increased sample sizes could impact the results, expanding clinical outcomes. Another limitation was a lack of access to baseline information on the health statuses of participants (i.e., medical records). The study participants described their overall health status by self-assessment. Possible underlying or confounding conditions were not excluded. The clinical groups, and criteria for inclusion into separate subgroups, were broad enough and limited only in general medical terms, which will be explored in further assessments and analyses. The statistical methods used in our analyses were limited to those that were deemed to be suitable for case-control studies. 

## 5. Conclusions

The results showed that self-isolation during COVID-19 pandemic quarantine affects people’s psychological health. The study showed that using psychological examination by PSS and laboratory stress indicators the results of the case group reliably differed from the results of the control group, concluding that self-isolation more often causes moderate chronic stress, with or without decompensation. The obtained research results expand the possibilities of using laboratory stress indicators in the daily work practice of a psychologist, showing the advantages of combining it with the psychological examination: assessing the PSS results—to determine the strength of stress, by laboratory stress markers—the duration of stressors or possible pathogenic causes. Besides the main objective additional value was found that laboratory stress biomarkers could be acceptable for broader clinical application during routine psychological counselling.

Our work revealed, the application of psychological scales with combined laboratory investigation could be a subject for further clinical studies assessing mental health status.

## Figures and Tables

**Figure 1 medicina-61-00175-f001:**
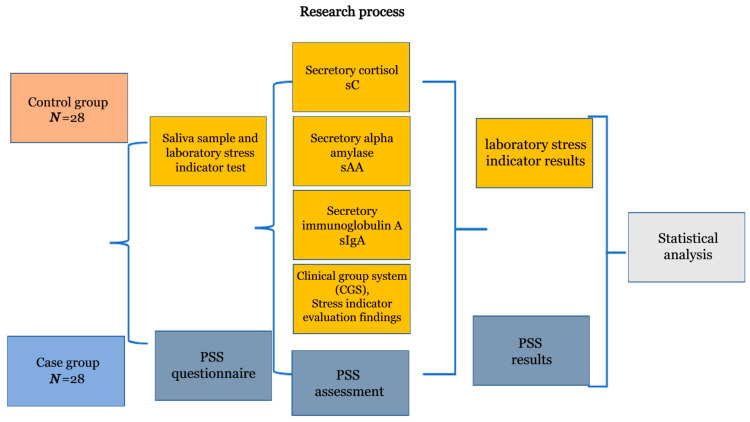
The course of the study.

**Figure 2 medicina-61-00175-f002:**
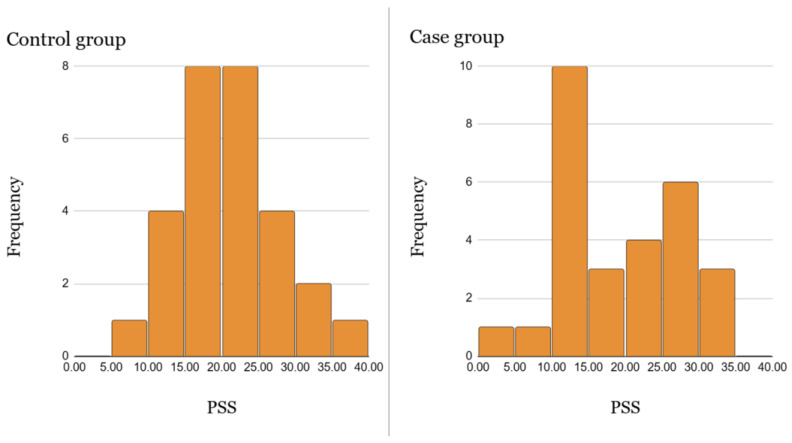
Percentage distribution of respondents in the case and control groups by frequency of outcomes in the groups.

**Figure 3 medicina-61-00175-f003:**
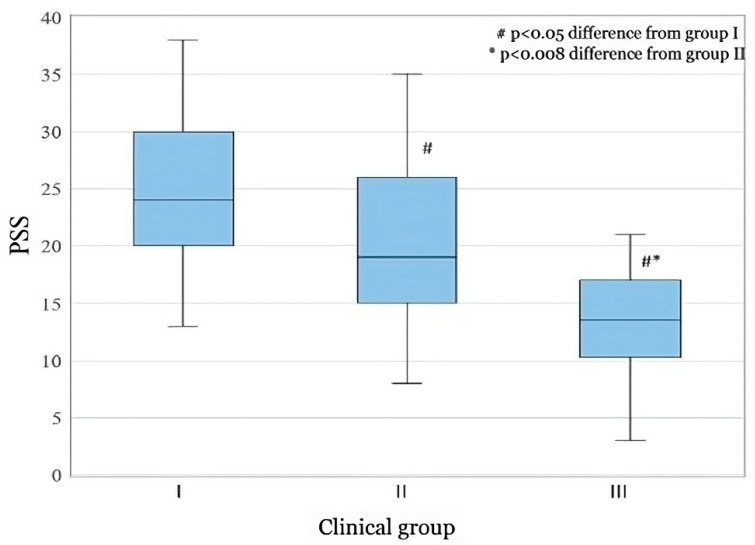
Distribution of mean PSS scores in the CGS groups for all respondents (*n* = 56).

**Table 1 medicina-61-00175-t001:** Comparison of the homogeneity of the case and control groups in terms of age and gender.

Criteria	Case Group	Control Group	Statistical Methods and *p* Value
Age, y.	22.0	22.0	Mann-Whitney
Median (min–max)	(18.0–24.0)	(18.0–24.0)	U = 381.5; *p* = 0.906
Gender			Chi squared test *χ*^2^ = 18.2, degree of freedom 1; *p* = 0.420
Male	14 (50%)	11 (39.3%)	
Female	14 (50%)	17 (60.7%)	

**Table 2 medicina-61-00175-t002:** Results of laboratory stress markers in the control and case groups.

Test Parameter/ Indicator	Control Group Median (Min–Max)	Experimental Group Median (Min–Max)	Mann–Whitney U. *p* Value
sC, nmol/L	10.85 (0.89–58.62)	17.23 (4.01–18.02)	U = 269; *p* = 0.044
sAA, kU/L	127.86 (10.12–500)	68.06 (10.54–500.00)	U = 315; *p* = 0.210
sIgA, mg/L	304.56 (44.37–1200.00)	829.18 (124.57–1200.00)	U = 170; *p* < 0.0001

**Table 3 medicina-61-00175-t003:** Correlations between the means of the indicators analyzed in the case group.

Case Group	Age	sC, nmol/L	sAA, kU/L	sIgA, mg/L	PSS
Age	-	0.093	0.892	0.740	0.258
sC, nmol/L	0.093	-	0.174	0.835	0.348
sAA, kU/L	0.892	0.174	-	R = 0.491 *p* = 0.008	0.348
sIgA, mg/L	0.740	0.835	R = 0.491 *p* = 0.008	-	0.387
PSS	0.258	0.314	0.348	0.387	-

**Table 4 medicina-61-00175-t004:** Correlations between the means of the indicators analyzed in the control groups.

Control Group	Age	sC, nmol/L	sAA, kU/L	sIgA, mg/L	PSS
Age	-	0.210	0.735	0.328	R = −0.400 *p* = 0.035
sC, nmol/L	0.210	-	R = −0.600 *p* = 0.024	R = 0.426 *p* = 0.024	0.883
sAA, kU/L	0.735	R = −0.600 *p* = 0.024	-	0.095	0.985
sIgA, mg/L	0.328	R = 0.426 *p* = 0.024	0.095	-	0.489
PSS	R = −0.400 *p* = 0.035	0.883	0.985	0.489	-

## Data Availability

Data used in this research can be provided by the corresponding author upon request.

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
