# Peer review of "Investigation of Perceived Stress During COVID-19 Pandemic Self-Isolation Periods"

_medicina, 2025, doi:10.3390/medicina61020175_

Round 1
Reviewer 1 Report
Comments and Suggestions for Authors
Overall, I feel that the study/topic is a little outside the current context. The pandemic has been going on for some time. Will be helpful to reformulate the introduction to be more fluid, coherent with actual context (it is to focus on the pandemic) topic of the article and to relate better with discussion section.
Limitations and future studies should be in the conclusion and not discussion.The conclusion is very concise and shortly.
Organization of the paper, grammar, and references: Another read through to check grammar errors and format would be helpful. Also review all references according with norms of the Journal.
Author Response
Dear Reviewer,
Many thanks for your revision and valuable comments addressed. Please, find our responses below. Added or amended sentences are highlighted in the manuscript.
1) Overall, I feel that the study/topic is a little outside the current context. The pandemic has been going on for some time. Will be helpful to reformulate the introduction to be more fluid, coherent with actual context (it is to focus on the pandemic) topic of the article and to relate better with discussion section.
Response: The pandemic allowed scientists to analyze the consequences and effects to mental health. Our study analyzed the differences between self-isolated and not affected groups. Amendments have been made in the text (highlighted), we have amended the title as well.
2) Limitations and future studies should be in the conclusion and not discussion. The conclusion is very concise and shortly.
Response: the conclusion and discussion are amended (highlighted).
3) Organization of the paper, grammar, and references: Another read through to check grammar errors and format would be helpful. Also review all references according with norms of the Journal.
Response: The manuscript has been revised, amendments have been made (highlighted).
Reviewer 2 Report
Comments and Suggestions for Authors
A case-control study conducted during the COVID-19 pandemic in Lithuania evaluates the associations between psychological and laboratory stress indicators (alpha-amylase, secretory cortisol, and immunoglobulin A) in the experimental and control groups. The topic covered by the authors of this manuscript is important from the point of view of public health and worth publishing. The obtained research results expand the possibilities of using laboratory stress indicators in the daily work practice of a psychologist.
In general, the manuscript is well written, coherent, and follows a logical development with good spelling and grammar and a good scientific style. In relation to the summary allows to understand the article without having to read it.The introduction properly and comprehensively introduces the issues raised by the authors of the article. The objectives of the study were correctly defined. The methodology is rigorous and well-described. The materials and methods describe the study in a comprehensive manner. Results correctly described, using tables and figures. Statistical analysis performed correctly. The discussion follows the same logical order as the results. The conclusions were formulated correctly.
Minor comments:
Materials and methods: Who gave the bioethical consent, and what date of consent? Was written consent to participate in the study obtained?
Line 132: “Respondents signed an informal consent agreement to participate in the study”.
Please explain the concept of informal consent
Line 379: Please indicate where the quote comes from
At the end of the Discussion, it is worth adding “Implications for the future”.
Author Response
Dear Reviewer,
Many thanks for your revision and valuable comments addressed. Please, find our responses below. Added or amended sentences are highlighted in the manuscript.
A case-control study conducted during the COVID-19 pandemic in Lithuania evaluates the associations between psychological and laboratory stress indicators (alpha-amylase, secretory cortisol, and immunoglobulin A) in the experimental and control groups. The topic covered by the authors of this manuscript is important from the point of view of public health and worth publishing. The obtained research results expand the possibilities of using laboratory stress indicators in the daily work practice of a psychologist.
In general, the manuscript is well written, coherent, and follows a logical development with good spelling and grammar and a good scientific style. In relation to the summary allows to understand the article without having to read it.
The introduction properly and comprehensively introduces the issues raised by the authors of the article. The objectives of the study were correctly defined. The methodology is rigorous and well-described. The materials and methods describe the study in a comprehensive manner. Results correctly described, using tables and figures. Statistical analysis performed correctly. The discussion follows the same logical order as the results. The conclusions were formulated correctly.
Minor comments:
1) Materials and methods: Who gave the bioethical consent, and what date of consent? Was written consent to participate in the study obtained?
Response: Bioethical Committee of Lithuania gave the bioethical consent, received in 31/01/2022 (added to the manuscript, in Materials and Methods section, highlighted). Informed consent was obtained from all subjects involved in the study in written form and signed, this sentence is added to the manuscript, highlighted (a copy has been added to the supplemental documents attached to the manuscript).
2) Line 132: “Respondents signed an informal consent agreement to participate in the study”. Please explain the concept of informal consent.
Response: The sentence is amended: Informed consent was obtained from all subjects involved in the study in written form and signed, highlighted in the text. The spelling error was missed.
3) Line 379: Please indicate where the quote comes from
Response: American Psychologist Association (APA) citated the co -author Julianne Holt-Lunstad, PhD, Professor of psychology and neuroscience at Brigham Young University of referenced Holt-Lunstad J et, 2015. We have amended this citation, highlighted in the text.
4) At the end of the Discussion, it is worth adding “Implications for the future”.
Response: Implications for the future are added and highlighted.
Reviewer 3 Report
Comments and Suggestions for Authors
Dear Editor,
Abstract:
1. The abbreviation PSS appears in the abstract but is only described in line 134.
2. There are ideas that are unclear, such as: "showed no objections between investigation methods."
3. The conclusion: "Study results suggest that laboratory stress biomarkers could be acceptable for broader clinical application during routine psychological counselling," does not align with the objective: "The study purpose was to analyze possible health consequences of self-isolation...
4. I recommend restructuring the abstract.
Introduction
1. It contains very long ideas supported by a single reference.
2. Line 34: "according to Johns Hopkins University data and sickening and infecting 34 hundreds of millions with SARS CoV2 virus," lacks a reference. Later, Wang is cited, but this paper does not provide information about the magnitude of COVID-19: https://www.mdpi.com/1660-4601/17/5/1729
3. Line 37: Wang's article does not show a direct relationship with the referenced idea.
4. Line 50: It is unnecessary to introduce an author being referenced, as in: “According to the meta-analysis co-authored by Julianne Holt-Lunstad, PhD, Professor of psychology and neuroscience at Brigham Young University.”
The introduction contains separate ideas that fail to converge. Therefore, I recommend restructuring it to establish a coherent narrative.
Materials and Methods
1. The type of study mentioned in the abstract and methods does not match (Case-Control and Experimental-Control are different).
2. The research process diagram does not provide additional information beyond what is described.
3. The presented figures have low definition (pixeled).
4. The explanation of the classification into different groups is overly superficial.
5. Line 140: Classifying test scores leads to a loss of information; I recommend conducting analyses as if they were based on raw counts.
6. The statistical analysis should account for other stress-generating conditions; in this case, it is essential to consider controlling for the confounding effects of multiple variables.
7. Making comparisons using "hypothesis tests" to draw conclusions overlooks the importance of the magnitude of the phenomenon.
Results:
1. Although the results align with those described, they should be approached from a different perspective.
For example: Cohen's effect size, among others.
Discussion:
1. Line 316 to 322: That paragraph appears to be a conjecture by the authors.
2. It contains very long paragraphs with only a single reference.
3. A greater number of limitations need to be described.
4. The conclusions described are not consistent with the limitations of the work.
General:
The terms "self-isolation" and "isolation" are used as if they were the same.
The study has a significant bias that necessitates careful interpretation of the conclusions:
1. The lack of a baseline makes it impossible to attribute the effects specifically to isolation.
2. Subjects who remained isolated for more than three months are likely to have an underlying medical condition (biological or mental) that leads to prolonged isolation. Having medical conditions inherently results in higher stress levels.
3. There is confusion in the taxonomy of the studies.
4. The statistical tests used have limitations that make it impossible to validate the results.
5. The papper title is unclear.
As a conclusión: I believe: this article has a series of limitations that prevent the proposed hypotheses from being resolved.
Author Response
Dear Reviewer,
Many thanks for your revision and valuable comments addressed. Please, find our responses below. Added or amended sentences are highlighted in the manuscript.
Abstract:
1. The abbreviation PSS appears in the abstract but is only described in line 134.
Response: Abbreviation PSS- Perceived Stress Scale is added and described in the abstract, the addition is highlighted.
2. There are ideas that are unclear, such as: "showed no objections between investigation methods."
Response: The ideas are clarified, amending the sentence, highlighted in the text. “between different investigation methods: psychological assessment by PSS and clinical laboratory testing using stress indicators”.
3. The conclusion: "Study results suggest that laboratory stress biomarkers could be acceptable for broader clinical application during routine psychological counselling," does not align with the objective: "The study purpose was to analyze possible health consequences of self-isolation...
Response: Conclusions are re written subsequently.
4. I recommend restructuring the abstract.
Response: The abstract is restructured.
Introduction
1. It contains very long ideas supported by a single reference.
Response: Restructured and shortened following up a coherent narrative. The parts which are removed are crossed out in the main text, added information is highlighted.
2. Line 34: "according to Johns Hopkins University data and sickening and infecting 34 hundreds of millions with SARS CoV2 virus," lacks a reference. Later, Wang is cited, but this paper does not provide information about the magnitude of COVID-19: https://www.mdpi.com/1660-4601/17/5/1729
Response: The reference is added and highlighted. An interactive web-based dashboard to track COVID-19 in real time. https://gisanddata.maps.arcgis.com/apps/opsdashboard/index.html#/bda7594740fd40299423467b48e9ecf6
3. Line 37: Wang's article does not show a direct relationship with the referenced idea.
Response: Explored explanation in the text related to the cited author (highlighted) “As the COVID-19 pandemic continued, various restrictions and bans continued globally despite the push for vaccination, leaving national decisions free to be more or less restrictive with subsequent disturbances of routine and stability. Provoked stressful conditions causing current psychological consequences to be a source for future investigations. (Wang et al, 2020).
4. Line 50: It is unnecessary to introduce an author being referenced, as in: “According to the meta-analysis co-authored by Julianne Holt-Lunstad, PhD, Professor of psychology and neuroscience at Brigham Young University.”
Response: Removed unnecessary information, amended in the text and highlighted: “According to the meta-analysis, a lack of social connections increases health risks as much as smoking 15 cigarettes a day or having an alcohol drinking problem. Scientists also found that loneliness and social isolation are twice as harmful to physical and mental health as obesity (Holt-Lunstad et al, 2015)”
5. The introduction contains separate ideas that fail to converge. Therefore, I recommend restructuring it to establish a coherent narrative.
Response: The introduction is re written to follow up a coherent narrative.
Materials and Methods
1. The type of study mentioned in the abstract and methods does not match (Case-Control and Experimental-Control are different).
Response: In our study we called the Experimental group by the same meaning as Case group according to the study type and purpose. Experimental group is re-named into Case group in the manuscript.
2. The research process diagram does not provide additional information beyond what is described.
Response: The research process diagram used with different colors for stages and design to visualize and emphasize two clinically separate methods used for the study. The idea was to show the summarized process for short presentation to the reader.
3. The presented figures have low definition (pixeled).
Response: Figures are revised, the definition now is acceptable.
4. The explanation of the classification into different groups is overly superficial.
Response: clarified and amended: Different clinical groups are classified according to Explanation of methodology taken from stress markers assays’ inserts being validated at Kaunas city Outpatient Clinic, added in the References and highlighted in the text:
Instructions for use. EUROIMMUN Medizinische Labordiagnostika AG, AlphaAmylase Saliva ELISA, assessed online in 12.01.2019.
Instructions for use. EUROIMMUN Medizinische Labordiagnostika AG, Cortisol Saliva ELISA, assessed online in 12.01.2019.
Instructions for use. EUROIMMUN Medizinische Labordiagnostika AG, Imunoglobulin A. Saliva ELISA, assessed online in 12.01.2019.
5. Line 140: Classifying test scores leads to a loss of information; I recommend conducting analyses as if they were based on raw counts.
Response: Study was designed, and results obtained for scores calculation according to known classification and the intervals of the scores by PSS validated methodology by Cohen (Cohen et al, 1983). The limitations are explored in discussion, highlighted.
6. The statistical analysis should account for other stress-generating conditions; in this case, it is essential to consider controlling for the confounding effects of multiple variables.
Response: We agree that lack of confounding effects of possible multiple variables, for example, other somatic symptoms or underlying conditions could play for broader vision and findings to be assessed. The respondents have been asked to state their basic status, taken medicines or any other information, related to additional effect or condition in the meantime filling the questionnaire. The limitations are discussed in Discussion, highlighted.
7. Making comparisons using "hypothesis tests" to draw conclusions overlooks the importance of the magnitude of the phenomenon.
Response: Possible effect of “hypothesis tests” haven’t been discussed as the study focused to the investigation of possible mental consequences during self-isolation caused by the “magnitude of the phenomenon” broadly described in Introduction.
Results:
1. Although the results align with those described, they should be approached from a different perspective.
For example: Cohen's effect size, among others.
Response: A very valuable comment. The sensitivity of significance testing to sample size is an important reason why many researchers advocate reporting effect sizes and confidence intervals alongside test statistics and p values. Statistical tools for effect size measurement, as Cohen’s effect d size, Pearson’s correlation or other as one of possible statistical significance methods are commonly used for quantitative Behavioral studies. Including effect size in quantitative analyses in particular, Cohen’s standard may help to evaluate the correlation coefficient to determine the strength of the relationship, or the effect size for our quantitative PSS results. Our point for this, getting back to the study results, to avoid fallacy estimation with very small sample size (28 individuals in each sample) that "effect size" for small samples is not the best to be calculated (Lakens, D., 2022. Sample size justification. Collabra: Psychology, 8(1), 33267). This assumption enables us to use the Statistics described in our Materials and Methods section.
Discussion:
1. Line 316 to 322: That paragraph appears to be a conjecture by the authors.
Response: Amended and highlighted.
2. It contains very long paragraphs with only a single reference.
Response: Revised and amended, highlighted. There are References which known the first one related to the single idea or emphasizes the main idea of the context in the Discussion.
3. A greater number of limitations need to be described.
Response: Limitations explored, and amendments are highlighted.
4. The conclusions described are not consistent with the limitations of the work.
Response: Amendments are added, and Conclusions are re-written.
General:
The terms "self-isolation" and "isolation" are used as if they were the same.
Response: revised, clarified and explored to be consistent focusing on self-isolation term, highlighted.
The study has a significant bias that necessitates careful interpretation of the conclusions:
1. The lack of a baseline makes it impossible to attribute the effects specifically to isolation.
Response: Conclusions and discussion are amended, highlighted.
2. Subjects who remained isolated for more than three months are likely to have an underlying medical condition (biological or mental) that leads to prolonged isolation. Having medical conditions inherently results in higher stress levels.
Response: Clarified in Methods and Materials section “The selection criteria for the case group were the population of Kaunas city, aged 18-24 years, who needed to be in self- isolation for Covid-19 and the time not longer than 3 months had been passed after their self-isolation. Added limitations in Discussion, highlighted.
3. There is confusion in the taxonomy of the studies.
Response: Amended taxonomy in whole text and figures.
4. The statistical tests used have limitations that make it impossible to validate the results.
Response: In order to calculate the results we used a combination of parametric and non-parametric tests, according to the distribution of the data, in order to draw accurate conclusions.
5. The papper title is unclear.
The title has been revised and made clear “Investigation of perceived stress during Covid 19 pandemic self-isolation periods”.
As a conclusión: I believe: this article has a series of limitations that prevent the proposed hypotheses from being resolved.
Response: Amendments added.